# Oats Lower Age-Related Systemic Chronic Inflammation (iAge) in Adults at Risk for Cardiovascular Disease

**DOI:** 10.3390/nu14214471

**Published:** 2022-10-25

**Authors:** El Hadji M. Dioum, Kevin L. Schneider, David J. Vigerust, Bryan D. Cox, YiFang Chu, Jeffery J. Zachwieja, David Furman

**Affiliations:** 1Quaker Oats Center of Excellence, PepsiCo Health and Nutrition Sciences, Chicago, IL 60607, USA; 2Edifice Health Inc., San Mateo, CA 94401, USA

**Keywords:** systemic chronic inflammation, biological age, oats, beta-glucan, polyphenols

## Abstract

Despite being largely preventable, cardiovascular disease (CVD) is still the leading cause of death globally. Recent studies suggest that the immune system, particularly a form of systemic chronic inflammation (SCI), is involved in the mechanisms leading to CVD; thus, targeting SCI may help prevent or delay the onset of CVD. In a recent placebo-controlled randomized clinical trial, an oat product providing 3 g of β-Glucan improved cholesterol low-density lipoprotein (LDL) levels and lowered cardiovascular risk in adults with borderline high cholesterol. Here, we conducted a secondary measurement of the serum samples to test whether the oat product has the potential to reduce SCI and improve other clinical outcomes related to healthy aging. We investigated the effects of the oat product on a novel metric for SCI called Inflammatory Age^®^ (iAge^®^), derived from the Stanford 1000 Immunomes Project. The iAge^®^ predicts multimorbidity, frailty, immune decline, premature cardiovascular aging, and all-cause mortality on a personalized level. A beneficial effect of the oat product was observed in subjects with elevated levels of iAge^®^ at baseline (>49.6 iAge^®^ years) as early as two weeks post-treatment. The rice control group did not show any significant change in iAge^®^. Interestingly, the effects of the oat product on iAge^®^ were largely driven by a decrease in the Eotaxin-1 protein, an aging-related chemokine, independent of a person’s gender, body mass index, or chronological age. Thus, we describe a novel anti-SCI role for oats that could have a major impact on functional, preventative, and personalized medicine.

## 1. Introduction

Despite being largely preventable, cardiovascular disease and related conditions are still the leading cause of death worldwide [1]. Recent evidence demonstrates that age-related systemic chronic inflammation (SCI) can have a profound effect on the aging of the cardiovascular system [2] and aging effects on other systems. For instance, SCI increases the risk of metabolic syndrome [2,3], type 2 diabetes [3], hypertension [4], cardiovascular disease [5,6], chronic kidney disease [5], cancer [7], depression [8], neurodegenerative and autoimmune diseases [9,10,11], and osteoporosis [12,13].

Age-related SCI is different from the canonical acute inflammatory response in several ways. Whereas acute inflammation is characterized by the activation of immune and non-immune cells that provide surveillance and protection against the full spectrum of microorganisms and toxic insults by eliminating them from the body and triggering tissue repair and recovery [14,15], SCI, or “inflammaging” [16], is a significant characteristic of the aging process and ensues in response to ‘sterile’ agents. The current understanding is that SCI is initiated by unresolved triggers of acute inflammation or physical, chemical, or metabolic noxious stimuli (i.e., “sterile” agents), released by damaged cells or environmental insults that are generally called damage-associated molecular patterns (DAMP) [4,16,17,18]. DAMPs promote a state of low-grade, systemic chronic inflammation characterized by the activation of immune components that are distinct from those triggered during an acute immune response [19,20,21]. Shifts in the temporal nature of the inflammatory response from the short- to long-term can result in significant immune deficit and lead to collateral damage to the organs and tissues. Not surprisingly, canonical biomarkers of acute inflammation such as Interleukin-6 (IL-6), Tumor Necrosis Factor-alpha (TNF-α), and high-sensitivity *C*-reactive protein (hs-CRP) may not be representative of the current chronic inflammatory load that the body may experience [22,23,24]. A novel metric for SCI based on multi-omics approaches was developed from the ten-year project across 1000 subjects at Stanford University called the 1000 Immunomes Project (1KIP) [25] and provided a personalized approach to determining the nature and quality of SCI in the individual participants. This metric (iAge^®^), derived from a deep learning algorithm applied to immune protein serum biomarkers, can predict multimorbidity, premature cardiovascular aging, immunological decline, frailty, and all-cause mortality. The biomarkers used to calculate the iAge^®^ and identified in the Stanford 1KIP study CCL11 (Eotaxin), Interferon-gamma (IFN-γ), Growth Regulated Oncogene-alpha (Gro-α), Monokine Induced by Gamma Interferon (CXCL9) and TNF-related Apoptosis-Inducing Ligand (TRAIL).

Chronic inflammation, inflammatory disease, and infection can induce a broad range of deficits in lipid metabolism, including decreases in serum (high-density lipoprotein) HDL, increases in triglycerides, lipoprotein a (Lp(a)), and low-density lipoprotein (LDL) [26]. The sustained levels of inflammation result from changes in lipid homeostasis, which is a major contributor to atherosclerosis risk [26,27]. In addition to affecting serum lipid levels, SCI can adversely affect lipoprotein function. For instance, the ability of HDL to prevent oxidation of LDL is severely diminished, and several steps in reverse non-HDL cholesterol efflux are also affected by the activation of the immune system [27]. It is now well established that soluble and cellular immune factors associated with SCI can promote inflammation-related endothelial dysfunction and atherogenesis [4,5,6,28]. Therefore, individuals with elevated iAge^®^ and LDL cholesterol levels represent a population at risk for cardiovascular disease (CVD) and other vascular complications.

Significant efforts have been made to identify methods for the modulation of SCI-mediated pathology. A chief example is the use of canakinumab (anti-Interleukin-1 (IL-1) antibody) to reduce cardiovascular events or lung-cancer-related mortality in patients with elevated inflammatory markers [29,30]. While these studies have provided substantial evidence regarding the modulation of biomarkers of SCI in the progression of age-related pathology, they were conducted in patients with existing disease and thus, could not address the effect of SCI modulators on disease prevention in otherwise clinically healthy yet, at risk, adults. 

Accumulating evidence from various clinical trials indicates that medical foods and natural compounds from food products can have substantial immunomodulatory effects. For instance, common oat (*Avena sativa*) and its constituents have gained considerable attention not only as important food supplements but also as immunomodulators with potential effects on the prevention of age-related disease [31,32,33,34,35,36]. Salient compounds found in oats that may mediate its clinical benefit include -glucans, as demonstrated by multiple studies in humans where supplementation with -glucan extract resulted in important health benefits, including a significant reduction in total cholesterol and low-density lipoprotein and CVD risk [37,38,39,40,41]. Furthermore, the US database ClinicalTrials.gov summarizes over 120 β-glucan clinical trials, mostly in cancer, metabolic syndrome, gastrointestinal tract therapy, lowering non-HDL cholesterol, and immune response improvements. In addition to β-glucans, oats contain multiple bioactive compounds, such as Fe, Zn, Se, and B- vitamin complex, all of which can positively impact immune system function [34]. Importantly, studies in animal models have repeatedly demonstrated that oats and its constituents can significantly lower several aging-related SCI proteins in both immune and endothelial cells [33,42,43,44]. In particular, avenanthramides and other polyphenols profoundly affect the expression of the age-related chemokine CCL11 [41], a key protein contributor to iAge^®^ and largely associated with cognitive capacity and resilience [45].

Here, we conducted secondary measurements using serum samples from a double-blinded placebo-controlled clinical study to investigate the effects of the consumption of an oat product (SoluOBC) providing 3 g of β-Glucan (OBG) compared to a rice control (PCB) after 2- and 4-weeks intervention (Appendix A) on blood lipids and cardiovascular disease risk biomarkers [39]. We used the serum samples to measure Eotaxin-1, and other SCI biomarkers and utilized the Inflammatory Age (iAge^®^) test (Edifice Health, Inc., San Mateo, CA, USA) as a study endpoint. We show that individuals with elevated baseline iAge^®^ and LDL cholesterol exhibit the greatest benefit from oats intervention.

## 2. Materials and Methods

A randomized, double-blind, placebo-controlled, parallel arm design clinical trial was performed at the contract research organization, INQUIS Clinical Research (Toronto, ON, Canada). The study was registered at www.clinicaltrials.gov with the identifier: NCT03911427.

### 2.1. Ethics

All experimental procedures using human subjects were followed in accordance with the ethical standards of the Western Institutional Review Board (IRB# 1256393) and, likewise, in accordance with the Helsinki Declaration of 1975, as revised in 1983.

### 2.2. Randomization and Concealment

Eligible participants were randomly assigned in a ratio of 1:1 to placebo or active treatment using blocks of various sizes to maintain balance among treatments while concealing treatment assignments from participants, study practitioners, and outcome assessors. Participants’ treatment assignments were based on the randomization sequence and in the order of their attendance at the baseline visit. Randomization was generated by a computer with random seed chosen from a table of random numbers and provided by the PepsiCo statistician [46].

### 2.3. Study Cohort and Treatment

A randomized, double-blinded, placebo-controlled interventional study was conducted on 191 healthy male and female patients (38% male) 21–65 years old (average age of 48) with moderate-to-elevated LDL-cholesterol levels (Appendix A). The study was conducted over four weeks with blood sampling every two weeks by standard venipuncture. Participants were recruited from the population of Toronto (ON, Canada) and surrounding metro areas using social media and other online advertisements, physical locations, and from the INQUIS database of volunteers who had given permission and expressed interest in future studies. Participant demographics and sample sizes are detailed in a previous publication [46]. Subjects consumed 3 sachets of the interventional oat product daily on 3 separate occasions to provide 3 g of OBC, as described previously [46] (*n* = 96 patients). The placebo group was given rice powder to match weight, energy (kcal), total fats, saturated fats, total carbohydrates, available carbohydrates, and protein concentration to the treatment and taken at the same frequency (*n* = 95 patients) (Appendix A). Serum samples (*n* = 573) were obtained from individuals at baseline, week 2, and week 4. Blood levels of Hemoglobin (g/L), Urea (mmol/L), Creatinine (umol/L), Gamma-glutamyl transferase (GGT) (U/L), Alanine transaminase (ALT) (U/L), Aspartate aminotransferase (AST) (U/L), % Saturated fatty acids (SFA), triglycerides (mmol/L), HDL (mmol/L), LDL (mmol/L), total cholesterol (mmol/L), fasting glucose (mmol/L), glycated albumin (mmol/L), insulin (mIU/L) were analyzed at baseline, 2 weeks and 4 weeks (LifeLabs, Inc. Mississauga, ON, USA). Blood pressure, heart rate, and weight were measured at each visit, and the patient answered several subjective questions. Serum samples were sent to the Edifice Health lab to analyze iAge^®^ modifiers and iAge^®^ determination. Study participants and serum samples from this group were previously studied in [46], and any adverse events from the study treatment are detailed in the Supplemental Data [46].

### 2.4. iAge^®^ Determination

Samples were analyzed using a Luminex LX-200 instrument (Luminex Corp., Austin, TX, USA) to determine the levels of Inflammatory Age^®^ markers using Edifice’s proprietary assay, composed of 5 core proteins CCL11, IFN-γ, GRO-α, CXCL9 and TRAIL. Raw mean fluorescent intensity (MFI) values for each plate below the 5th percentile were set to the 5th percentile of the plate and those above the 95th percentile were set to the 95th percentile of the plate. These values were normalized using control serum samples from eleven individuals spanning a diverse range of ages (23–83 years old) and both sexes. iAge^®^ was derived from all study participants using the Edifice proprietary machine learning algorithms using the normalized MFI values. The SCI index is calculated for each individual from the empirical cumulative distribution of iAge^®^ in the study population from the same decade as the individual.

### 2.5. Statistical Analysis

Samples were grouped by treatment. Changes in the protein modulators of iAge^®^ or on iAge^®^ at week 2 vs. baseline and week 4 vs. week 2 were compared using a one-tailed pairwise *t*-test. The least absolute shrinkage and selection operator (LASSO) machine learning algorithm was implemented using the elastic net [8] module in R program (RStudio, Boston, MA, USA) and used to train a model on age, sex, ethnicity, and other measured baseline traits to predict responders to treatment in the oats and control groups. The predictive power of the known and measured attributes in this model is derived from the variable coefficients of the LASSO analysis. Responders were defined as the difference of iAge^®^ at week 2 from baseline, starting at zero and decreasing by one until the area under the curve no longer increased. Similarly, those with high iAge^®^ at baseline were used to calculate the optimal cut point analysis for the responders to the oat treatment. The variable with the largest coefficient in the LASSO model (baseline iAge^®^, see below) was used to calculate the optimal cut point that individuals are responders to oat treatment using cutpointR (package by Christian Thiele at github.com/Thie1e/cutpointr) in R [9]. 

## 3. Results

### 3.1. High Baseline iAge^®^ Predicts the Effectiveness of Oats Intervention

Study participants in the OBG group did not display any significant difference in their calendar Age, iAge^®^ and serum Eotaxin-1 levels, compared to PCB control at baseline (Appendix A). Compared to baseline, the iAge^®^ at 2 weeks post intervention showed a trend toward a decrease in iAge^®^ in the oat treatment (OBG) group, although this was not statistically significant. In comparison, the iAge^®^ in the rice control group (PCB) did not show any significant variation (Appendix A). To examine whether baseline attributes can predict changes in iAge and its proteins at week 2 and week 4 following daily consumption of OBC or PCB, we applied Least Absolute Shrinkage and Selection Operator (LASSO) regression, a standard machine learning approach largely utilized for predictive tasks and feature selection. LASSO imposes a penalty to the regression coefficients such that small coefficients are shrunken to zero, and only the most relevant features are selected. The changes in iAge^®^ for each subject in the study were calculated. A total of 39 baseline features, including the subject’s demographics, blood biomarkers for cardiovascular health and lipid fractions, liver function, metabolism, and inflammation, were used as input (predictor) variables (see Methods). Using this method, four variables were selected for the prediction of the changes in iAge^®^ in week 2 versus baseline in the OBC group, and these provided a relatively robust prediction accuracy (cross-validated area under the curve (AUC), cvAUC = 0.72) (Figure 1A). Of the four baseline predictors selected by our model (Figure 1B), two were negative predictors, and these include total fiber content (g/day) and systolic blood pressure, and two were positive, including the percentage of dietary saturated fatty acid intake (%SFA) and baseline iAge^®^. Of these four features, baseline iAge^®^ was the strongest predictor of response to the oat product intervention (Figure 1B). The model for predicting changes in iAge^®^ using baseline parameters in the PCB group yielded a cvAUC = 0.48 (no different from a random distribution), suggesting that no relevant features contribute to the stratification of responders vs. non-responders in this group (Figure 1C).

Elevated iAge^®^ and circulating LDL have been shown to independently contribute to accelerated cardiovascular pathology [45,47]. Thus, to explore the effects of baseline iAge^®^ and LDL in response to oat treatment, an unbiased approach (cutpointR, see Methods) divided the cohort into those with low and high baseline iAge^®^ with a change in iAge^®^ at week 2 as the output variable (most changes were observed at the 2 weeks timepoint). The optimal separation of responders and non-responders to the oat product OBC was observed at an iAge^®^ value of 49.6 years. Individuals with an iAge^®^ above or below 49.6 were classified as “high” and “low” iAge^®^ groups, respectively. The high iAge^®^ group exhibited a significant decrease in iAge^®^ at week 2 (−1.46 iAge years, *p* = 0.008), whereas no significant change was observed in the low iAge^®^ group (Figure 2A,B). No significant change in iAge^®^ or its proteins was observed in the low or high iAge^®^ groups of the PCB group (Figure 2C,D). Similarly, the high iAge^®^ group in the OBC group showed a significant decrease in CCL11 at week 2 (−7% change, *p* = 0.002), but no significant decrease was observed in the low iAge^®^ group (Figure 2E,F). No significant change in CCL11 was observed in either the “low” or the “high” iAge^®^ groups within the PCB subjects (Figure 2G,H).

Next, to examine the effect of baseline LDL on further subject subsetting within the responder (high baseline iAge^®^) and non-responder (low baseline iAge^®^) groups, we applied the same method as before (cutpointR) with circulating baseline LDL levels as the input variable. The most powerful cutoff for baseline LDL levels was 3.27 mmol/L to predict response to OBC in patients with high baseline iAge^®^. For individuals with high baseline iAge and high LDL levels, there was a significant decrease in iAge^®^ at week 2 compared to baseline for the OBC-treated group (−2.3 years, *p* = 0.0027). The PCB control group showed no significant change (*p* = 0.27) (Figure 3) regardless of baseline iAge^®^ and LDL levels. 

These results indicate that oats can have an impact on iAge and CCL11 in subjects at risk for CVD with elevated baseline iAge and LDL cholesterol. 

### 3.2. Changes in iAge^®^ Induced by the Oat Product Are Correlated with a Decrease in CCL11

To unbiasedly investigate which iAge^®^ core proteins were targets for the oat product treatment and likely drove the modifications in iAge^®^ observed in this study, we examined the changes in iAge^®^ induced by the oat product treatment at week 2 and compared them with those observed in iAge^®^ proteins. In subjects with elevated baseline iAge^®^ (>49.6 iAge^®^ years), decreasing circulating levels of CCL11 tracked with changes at week 2 observed in iAge^®^ following the oat product treatment (Figure 4A, blue). No significant decreases in both CCL11 and iAge^®^ were observed in subjects with normal to low baseline iAge^®^ (<49.6 iAge^®^ years) in the OBC group (Figure 4A, gray). No significant changes were observed between iAge^®^ and any other iAge^®^ core protein in subjects with elevated or normal to low baseline iAge^®^. Similarly, no changes were detected in either the low or the high baseline iAge^®^ groups in the PCB cohort (Figure 4B). 

Together, these results indicate that the improvements in iAge^®^ observed in the oats group may be driven by the changes observed in CCL11 expression, and thus compounds found in oats may specifically target CCL11, as previously demonstrated in animal models [42].

## 4. Discussion

Here, we show that in subjects at risk for cardiovascular pathology, oats have a beneficial effect in lowering risk factors such as iAge and LDL cholesterol. With a total economic toll on our health care system of USD 216 billion per year and causing USD 147 billion in lost productivity on the job [48], our results show that oats can be a safe and accessible medicinal food with the potential to mitigate CVD risk and ameliorate the economic burden in our society.

The role of SCI in CVD has been repeatedly shown in numerous studies. For instance, our group demonstrated that the increased expression of inflammasome genes, including the flagellin sensor Nod-like receptor (NLR) family caspase recruitment domain (CARD)-containing protein 4 (NLRC4) and IL-1β are strong predictors of hypertension and arterial stiffness [4]. More recently, using an unbiased approach to immunity and aging, we developed the iAge^®^ metric, a first-in-class SCI tool that provides knowledge for scientific evidence-based interventions and treatments that may alter the trajectory of chronic disease management [47]. We reported that iAge^®^ is an important determinant of multimorbidity, longevity, frailty, immune aging, and all-cause mortality. The compounds found in oats can have a direct effect on iAge^®^ and CCL11, and therefore these have the potential to decrease the burden of age-related disease, be utilized in preventative medicine, and profoundly impact healthcare systems.

As predicted, the changes observed in iAge^®^ were due, in part, to a down-regulation in the inflammatory biomarker CCL11, in agreement with studies in mice showing that the bioactive compounds found in oats can lower the expression of this and other canonical age-related inflammatory proteins [4,49,50,51,52,53] such as Interleukin-12 (IL-12)p40, IL-1β, Macrophage inflammatory protein-1 alpha (MIP-1α), and Monocyte chemoattractant protein-1 (MCP-1) [42]. The changes in GRO-α observed in the control group did not significantly affect the iAge (Appendix A). Our work and others also suggest that oats have significant anti-inflammatory activity and the ability to reduce LDL [46]. In the pathogenesis of the cardiovascular disease, this double effect may benefit those in high-risk categories. The potential reduction in endothelial permeability derived from the downregulation of CCL11 may prevent LDL and oxidized LDL (ox-LDL) from moving into the vessel intima. That activity, coupled with the oats’ ability to sequester and eliminate lipids from the GI tract, may overall reduce serum lipid concentrations and the ability of LDL/ox-LDL to promote atherosclerosis. 

The effect of oats on the inflammaging-related chemokine CCL11 may have multiple implications for general health. For instance, this immune protein, largely produced by eosinophils, has traditionally been described as playing an essential role in allergic conditions. Beyond the contribution to allergic conditions, CCL11 has been shown to impact the permeability of vascular endothelium in a dose-dependent manner [51]. Niccoli et al. have suggested that eosinophils may play a role in coronary atherosclerotic disease, possibly through the expression of CCL11 [52,53]. Since the eosinophil is one of the primary secretors of CCL11, this further supports the idea that the downregulation of CCL11 may improve cardiovascular disease risk. In addition to the role of eosinophils in the expression of CCL11, there is evidence that macrophages and cardiac fibroblasts are also important expressers of CCL11, leading to the observed enhanced permeability of the human coronary artery endothelium [51]. Not only do macrophages express CCL11, but they also capture and clear oxidized LDL via scavenger receptors and CD36 on their membranes [54]. More recently, CCL11 has been demonstrated to have a role in brain disorders and as an emerging molecular signature of aging [45]. As a marker of aging, CCL11 has also been shown to have associations with neuroinflammation, neurodegeneration, and psychiatric disorders [45]. Our studies suggest that the ingestion of oats lowers the expression of CCL11 and could thereby decrease the effects of aging on the participants. 

The decreases in LDL (observed in previous studies of the oat product [37,38]), CCL11, and iAge^®^ may be due to the higher concentration and activity of polyphenols in the oat product along with β-glucans. In particular, participants in the treatment group ingested 3 g of β-glucan, 46.8 mg of total phenolics, and 1.14 mg of avenanthramides (A, B, and C) daily, which were not present in the rice powder given to the placebo group. Each of these components may directly or indirectly impact CCL11 and iAge^®^. For example, β-glucans in oat extract can increase microbiome diversity, as previously observed [55,56,57], and this could result in a decreased translocation of gut luminal antigens to the circulation, which, in turn, may result in an overall reduction in inflammatory markers 

Similarly, as is seen in cardiovascular disease and atherosclerosis, there is evidence that high levels of LDL and total cholesterol are risk factors for central nervous system pathophysiology [58,59,60]. Interestingly, no significant associations were found between iAge vs. LDL or HDL at baseline (Appendix A). Previous studies on OBC have demonstrated a direct effect on LDL and cholesterol metabolism, and could thus positively influence both cardiovascular disease and neurological disorders such as Alzheimer’s Disease and Parkinson’s Disease [60,61]). More generally, it has been well established that oats and their constituents have a beneficial effect on various health-related mechanisms, including cholesterol maintenance, satiety, and protection from carcinogenesis in the colon [62,63,64]. Oats have greater benefits in lowering lipid concentration than any other grain; various bioactive compounds and significant levels of dietary fiber in the form of β-glucans that all contribute to improved gut health [56]. The mechanisms that may be important in the ability of dietary fibers such as β-glucans to improve gut health and overall inflammatory health may be several-fold [34]. First, the ability of the gut to ferment these dietary fibers may allow for mucus-sparing and promote a stable and beneficial physical barrier. Secondly, the components in oats can stimulate and maintain a healthy and diverse gut microbiota. The consumption of oats has been shown to increase gut microbial diversity, leading to a significant increase in *Bifidobacteria* and *Lactobacilli* [63]. Thirdly, the fiber found in oats is metabolized by gut microbes such as *Bifidobacteria* into short-chain fatty acids (SCFA) that stimulate mucus, promote the expression of anti-microbial peptides, and, importantly, increase the expression of tight junctions. Collectively, these mechanisms support healthy microbial activity and promote chemical and physical barriers to the translocation of microbes and materials from the gut lumen into the bloodstream. Compared to other wholegrain food sources, oats are especially healthy due to their high amount of complex lipids, phenolics, and dietary fiber [56]. 

This study supports that those compounds found in oats have additional benefits in decreasing age-related systemic chronic inflammation in at-risk patients, which could significantly impact functional and preventative precision medicine.

## 5. Conclusions

Cardiovascular disease is the leading cause of death worldwide. Despite the well described effects of LDL-cholesterol in the development of this pathology, emerging evidence demonstrates that age-related SCI also plays an important role. The oat is a species of cereal grain rich in fiber that has the potential to lower non-HDL cholesterol levels, and recent evidence demonstrates that it can lower the levels of inflammatory mediators in animal models. Here, we demonstrate that an oat beverage providing 3 g of the soluble fiber β daily can lower SCI and CVD risk in human subjects with elevated baseline LDL-cholesterol and SCI. Thus, oats can have a beneficial effect on the inflammatory system and cardiovascular disease prevention in a subset of subjects presenting elevated risk factors. Additional studies focusing on that target population will enable a generalization of these results and optimization of an oat-based nutritional intervention to promote healthy aging and longevity.

## Figures and Tables

**Figure 1 nutrients-14-04471-f001:**
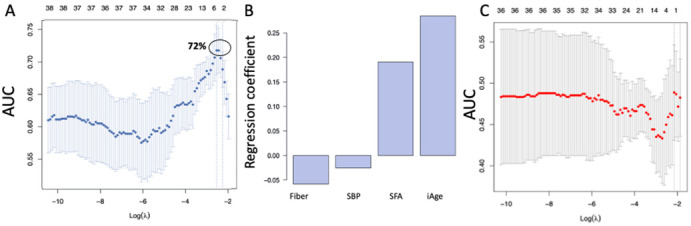
LASSO modeling to identify baseline features that correlate with the change in iAge^®^ in response to oat product. (**A**) Using the measured baseline traits of individuals in the oats group, LASSO classifier models were trained using a binary outcome of decreased/improvement (1) or increased/worsening (0) iAge^®^. Several LASSO models were evaluated with 3-fold cross-validation over varying lambda penalties. The mean accuracy of each lambda-penalized LASSO classifier model, given by the cross-validation area under the curve (cvAUC), was plotted along the y-axis, and the lambda penalty for each model is the x-axis. The indicated best performing LASSO classifier model shows a cvAUC = 0.72. (**B**) The coefficients from the best performing LASSO model from (**A**) were extracted showing the contributions of different metrics towards the prediction of decreased iAge^®^. The extracted coefficient values (y-axis) for the four features (x-axis) are plotted in this bar graph. Larger magnitudes of the coefficients (either positive or negative) represent larger roles in dictating the outcome of the LASSO model. (**C**) Multiple LASSO models with different lambda penalties were tested for the placebo-treated control group with 3-fold cross-validation. None of these LASSO models trained on the placebo group delivered a cvAUC score above random (cvAUC > 0.50). iAge^®^, Inflammatory Age^®^; SBP, Systolic Blood Pressure; SFA, saturated fatty acid.

**Figure 2 nutrients-14-04471-f002:**
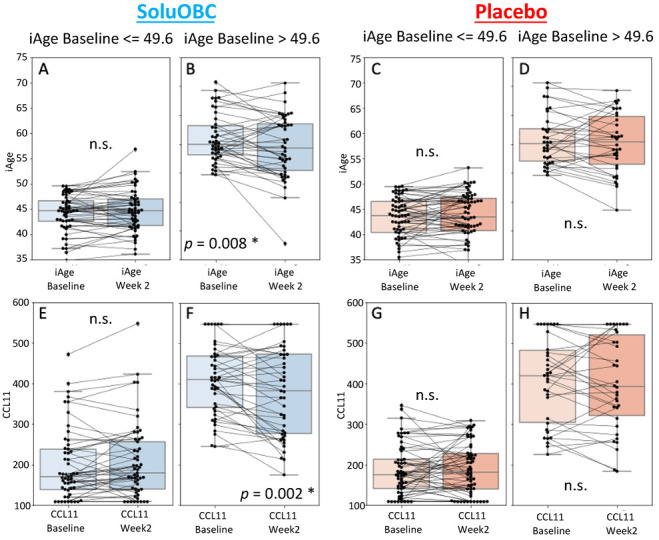
Effects of stratification by baseline iAge^®^ on changes in iAge^®^ and CCL11 after 2 weeks of Oats treatment. The value for iAge^®^ or CCL11 (y-axes) is provided for each patient as a black dot, and the change in that value for each patient between baseline as week 2 (x-axes) is represented by a connecting black line. The statistics for each group are shown using box plots, and significance between baseline and week 2 was calculated using a pairwise two-tailed *t*-test. The optimal cut-point for baseline iAge^®^ was calculated to be 49.6 years using cutpointR. Stratification of individuals treated with oats by baseline iAge^®^ showed no significant (n.s.) change in iAge^®^ at week 2 for those with a low baseline iAge^®^ (**A**), with a significant (asterisk) decrease for those with a high baseline iAge^®^ (**B**). No change was observed in iAge^®^ at week 2 with the placebo group (**C**,**D**). Stratification of individuals treated with oats by baseline iAge^®^ showed no significant change in CCL11 at week 2 for those with a low baseline iAge^®^ (**E**) and a significant decrease for those with a high baseline iAge^®^ (**F**). No change was observed in iAge^®^ at week 2 with the placebo group (**G**,**H**). CCL11, Eotaxin; SoluOBC, oat product providing 3 g of β-Glucan.

**Figure 3 nutrients-14-04471-f003:**
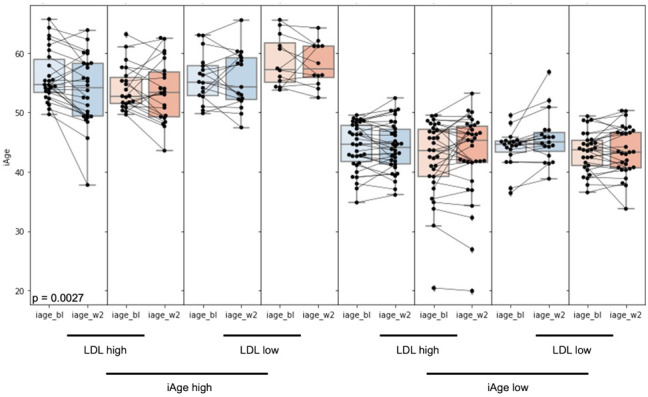
Effects of stratification by both baseline iAge^®^ and LDL levels on the change in iAge^®^. The value for iAge^®^ (y-axes) is provided for each patient as a black dot, and the change in that value for each patient between baseline as week 2 (x-axes) is represented by a connecting black line. The statistics for each group are shown using box plots, and significance between baseline and week 2 was calculated using a pairwise two-tailed *t*-test. The oat product (blue) and placebo (red) treated groups were subset into high/low baseline iAge^®^ using a cutoff of 49.6 years and high/low baseline LDL levels using a cutoff of 3.27 mmol/L. The only significant change in iAge^®^ at week 2 was observed with the oat treated group that had both high baseline iAge^®^ and LDL levels. LDL, low-density lipoprotein.

**Figure 4 nutrients-14-04471-f004:**
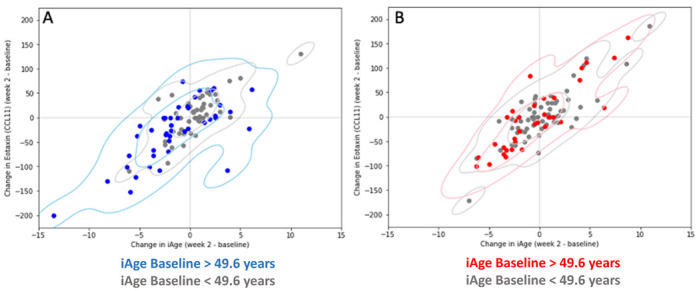
Effects of CCL11 changes on iAge^®^. The change in iAge^®^ between baseline and week 2 (x-axis) is compared to the change in CCL11 between baseline and week 2 (y-axis) for each patient. Individuals were stratified based on a high baseline iAge^®^ (blue circles) or low baseline iAge^®^ (grey circles). (**A**) Those in the oat group and have a high baseline iAge^®^ trend towards a lower iAge^®^ at week 2 and lower CCL11 level. (**B**) Those in the placebo group have no decreasing or increasing iAge^®^ or CCL11 level trends.

## Data Availability

Not applicable.

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
