# Peer review of "Oats Lower Age-Related Systemic Chronic Inflammation (iAge) in Adults at Risk for Cardiovascular Disease"

_nutrients, 2022, doi:10.3390/nu14214471_

Round 1
Reviewer 1 Report
Dear Editor,
I carefully read the manuscript by Dioum et al.
My comments and suggestions for the authors are the following:
- Do the authors refer either to HDL and LDL particles or cholesterol concentrations, throughout the manuscript? This is indeed a critical issue.
- Line 158: Why did the authors consider as statistically significant a one-tailed p-valus? I warmly suggest them to repeat the analysis by considering as statistically significant a p-value< 0.05!
- Line 159: Did the authors perform the Levene's test befor the Student's T test? How was the normal distribution of the variables tested, a priori?
- Did the authors follow the CONSORT guidelines? I warmly suggest them to include the CONSORT flow-chart in the results paragraph of the paper.
- The authors should consider to refer to PMID: 32138344
Author Response
Thank you for the valuable suggestions and comments. I updated the manuscript to reflect the changes.

Reviewer 2 Report
The paper needs significant improvement in the way that the methodology and the results are presented. Suggest that the authors download and use the CONSORT 2010 checklist (Consolidated Standards of Reporting Trials) as the basis for what should be reported in this clinical trial or for how the paper should be written. https://www.equator-network.org/reporting-guidelines/consort/

Author Response
Thank you for the valuable comments and recommendations. I hope the new version of the manuscript is an improvement that addressed your suggestions.

Round 2
Reviewer 1 Report
Dear Editor,
I carefully read the revised version of the manuscript that is significantly improved. I suggest to publish it in the Journal.
Author Response
I appreciate the diligence and insights you provided to improve the quality of the manuscript significantly. Thank you for the comments and suggestions. I updated the small corrections as recommended.
Thank you
Reviewer 2 Report
The paper has significantly improved since the last submission. The authors just need to improve on some of the revised sentences as there are some grammar issues and there is a hanging sentence in the abstract as follows:
" In a recent placebo controlled randomized clinical trial, an Oat product proving [providng ] 3 g of β-Glucan improved cholesterol LDL levels and lowered cardiovascular risk in adults with borderline high cholesterol. Here, we conducted a secondary measurement of 14 the serum samples to test whether the oat product has the potential to reduce SCI and improve other clinical [parameters? related to SCI. ]"
Other than this correction, the article is already acceptable for publication.
Author Response
Thank you for reviewing the manuscript and for providing valuable suggestions to improve the paper.